# Genome-Wide Identification and Expression Patterns of the *SWEET* Gene Family in *Bletilla striata* and its Responses to Low Temperature and Oxidative Stress

**DOI:** 10.3390/ijms231710057

**Published:** 2022-09-02

**Authors:** Chan Lu, Jun Ye, Yuanqing Chang, Zeyuan Mi, Shuai Liu, Donghao Wang, Zhezhi Wang, Junfeng Niu

**Affiliations:** National Engineering Laboratory for Resource Development of Endangered Crude Drugs in Northwest China, Key Laboratory of the Ministry of Education for Medicinal Resources and Natural Pharmaceutical Chemistry, College of Life Sciences, Shaanxi Normal University, Xi’an 710119, China

**Keywords:** *Bletilla striata*, expression pattern, subcellular localization, *SWEET*

## Abstract

SWEETs (sugars will eventually be exported transporters), a well-known class of sugar transporters, are involved in plant growth and development, sugar transport, biotic and abiotic stresses, etc. However, to date, there have been few investigations of *SWEETs* in Orchidaceae. In this study, 23 *SWEET* genes were identified in *Bletilla striata* for the first time, with an MtN3/saliva conserved domain, and were divided into four subgroups by phylogenetic tree. The same subfamily members had similar gene structures and motifs. Multiple *cis*-elements related to sugar and environmental stresses were found in the promoter region. Further, 21 genes were localized on 11 chromosomes and 2 paralogous pairs were found via intraspecific collinearity analysis. Expression profiling results showed that *BsSWEETs* were tissue-specific. It also revealed that *BsSWEET10* and *BsSWEET18* were responsive to low temperature and oxidative stresses. In addition, subcellular localization study indicated that *BsSWEET15* and *BsSWEET16* were localized in the cell membrane. This study provided important clues for the in-depth elucidation of the sugar transport mechanism of *BsSWEET* genes and their functional roles in response to abiotic stresses.

## 1. Introduction

Sugar is an essential organic substance and important source of energy for plants [1], in which glucose, sucrose, etc., are soluble sugars, whereas starch and cellulose are insoluble sugars [2]. These sugars have a variety of functions, such as providing carbon sources for other substances, regulating osmotic pressure, and providing energy for vital cellular activities [3,4,5,6,7].

Plants use photosynthesis to convert atmospheric CO_2_ to sources of sugar that they can use [8]. Mesophyll tissues that synthesize sugars in plants are called source tissues, whereas others that cannot are referred to as sink tissues. The sugars synthesized in source tissues are transported and assigned to sink tissues to ensure normal plant growth and development [9]. In higher plants, sucrose is synthesized via photosynthesis in mesophyll cells (source tissues), and transported over long distances by sugar transporters to plant organs (sink tissues), such as roots, stems, flowers, and seeds, providing nutrients for the growth and development of new cells [10,11,12]. Since sugar compounds, (e.g., sucrose, glucose, and fructose) cannot be transported to sedimentary tissues through plant biofilm systems they require the assistance of corresponding sugar transporters, which are key to their distribution [13,14]. SWEET (sugars will eventually be exported transporter) comprises a notable class of sugar transporter that was discovered in plants in recent years [15], which, independently of the environmental pH, could promote the diffusion of sucrose to the plasmodesmata along the concentration gradient during efflux. Further, it can regulate the uptake of glucose through the cell membrane in the form of glucose transporters, with the function of bidirectional sugar transporters [16]. A typical SWEET protein consists of seven transmembrane domains (TMs), of which three are more conserved TMs that form the MtN3/saliva structural domain, and the middle is connected by a less conserved TM [17]. Studies have shown that SWEET proteins can homo- and heterooligomerize to create functional pores; thus, providing transport channels and driving their functions [18].

SWEET proteins have been shown to affect important physiological processes during plant growth and development by regulating the transport, distribution, and storage of sugar compounds [19]. SWEET proteins were observed to be involved in phloem loading during plant sugar metabolism, where *AtSWEET11* and *AtSWEET12* were not only localized in the cytoplasmic membrane, but also highly expressed in the cytoplasms of thin-walled phloem tissues, which could sequester the sucrose of leaves into vascular bundles [20]. SWEET proteins were also associated with the development of flowers, fruits, and seeds, where *AtSWEET8* primarily affected fertility during the early development of inflorescence, and *AtSWEET13* mainly influenced fertility in later inflorescence development [21,22]. Meanwhile, the *SWEET* gene family is involved in plant–pathogen interactions, such as the sugars produced by *OsSWEET11*/*14*, which provide nutrients for *Xoo* (*Xanthomonas oryzae* pv. *oryzae*) and stimulate its growth and development [23,24]. Further, the involvement of SWEET warrants special attention in the development of root and stem shoots. Previous studies indicated that the formation of potato tubers depended on the relationship between the photoperiod and sugar transport. *StSP6A* interacts with *StSWEET11* to prevent sucrose leakage and promote symbiotic sucrose transport to regulate potato tuber formation [25], which also opens the possibility of source–sink interactions.

*Bletilla striata* (Thunb.) Reichb. f. is a perennial herb of the Orchidaceae family, which is not only an ornamental plant in Europe and America but also a significant astringent medicinal plant that is native to East Asia, where its dried pseudobulb is used as medicine [26,27,28]. Polysaccharides and stilbenes are the main bioactive chemical components of the genus *Bletilla*, and modern pharmacological studies show that there are hemostatic, anti-inflammatory, anti-ulcer, anti-bacterial, and anti-tumor pharmacological activities [29]. It is well-known that metals such as Cu have the potential to generate •OH and induce oxidative stress [30]. *BsPP2C*, *BsSnPK2.6,* and *BsPYL6* responded to various stresses, including low temperature and oxidative stress [31]. The PSII photosynthetic system of *B. striata* is disrupted by low temperature treatment, which further affects plant growth [32]. Due to its important medicinal value and significantly reduced wild natural resources, researchers have begun to cultivate it artificially [33], being mindful that polysaccharides play a crucial role in its tuber quality. Although *SWEET* genes have been functionally characterized in *Arabidopsis thaliana* and other plants, their functions in most species remain unclear. This is particularly the case for *B*. *striata*, for which the *SWEET* gene family remains poorly understood.

Since gene functional analysis is the core pursuit of plant molecular biology, this study aimed to conduct a genome-wide analysis of the *SWEET* gene family in *B. striata*, with a specific focus on physicochemical properties, phylogenetic relationship, chromosomal location, conserved motifs, gene structures, *cis*-acting elements in the promoter region, and synteny analysis. We also investigated the expression patterns of *BsSWEET* genes in different tissues under low temperature and oxidative stress. Additionally, we performed subcellular localization study to further explore whether *BsSWEETs* were involved in sugar transport. This study will provide insights for future investigations into plant stress response. Further, it will lay the foundation for research into the regulation of *SWEET* genes toward optimizing the accumulation and distribution of sugar.

The *SWEET* gene family has not been systematically and completely analyzed in medicinal Orchidaceae. This is the first time we performed a comprehensive analysis of the *SWEET* gene family in *B. striata*, expression patterns under different treatments, and the subcellular localization of *BsSWEET15*/*16*. Finally, we will discuss the possibility that *BsSWEET15*/*16* may form homodimers or heterodimers, and speculate that *BsSWEET15* and *CTG2800.4* may interact in sucrose transport in *B. striata*. New insights will be provided for the study of *SWEET* genes related to growth and development and sugar transport in medicinal Orchidaceae.

## 2. Results

### 2.1. Identification and Characterization of BsSWEET Genes

A total of 23 *BsSWEET* genes were identified based on gene annotation data from NR and KEGG databases and a comparison of conserved domains. SMART, InterPro, and NCBI-CDD revealed that all *BsSWEETs* contained one or two MtN3/saliva conserved domains (PF03083). We designated them *BsSWEET1*~*BsSWEET23* according to chromosomal location (Appendix A).

The physicochemical characteristics of the *BsSWEET* genes (e.g., name, protein length, molecular weight, and isoelectric point) are shown in Appendix A. The number of amino acids contained in BsSWEET proteins ranged from 112 (BsSWEET9) to 299 aa (BsSWEET16), whereas the molecular weights ranged from 12.5 kD to 33.4 kD. Except for BsSWEET5 and BsSWEET9, the isoelectric points of other BsSWEET proteins were greater than seven; thus, it was presumed that most BsSWEET proteins were basic proteins. Moreover, all BsSWEET proteins were hydrophobic and had no signal peptides (Appendix A). The subcellular localization prediction indicated that all SWEET proteins were located in the plasma membrane. Most BsSWEET proteins typically contained six to seven transmembrane domains, and the transmembrane domain structure was dominated by α-helix (Appendix A). It was speculated that SWEET proteins provided sugar transport channels in the plasma membrane. The abundance of phosphorylation sites in the BsSWEET proteins indicated that they were regulated by kinases that induced functions through phosphorylation and dephosphorylation, or might have been co-regulated by other regulatory modes (Appendix A).

### 2.2. Multiple Sequence Alignment and Phylogenetic Relationship of BsSWEET Genes

To obtain an in-depth understanding of the characteristics of the conserved *BsSWEET* domains, we conducted multiple sequence alignments and found that most *BsSWEET* gene family members had at least one complete MtN3/saliva domain, among which *BsSWEET2*/*5*/*9*/*12*/*13*/*19* contained only one MtN3/saliva conserved domain, whereas the other *BsSWEET* genes had two (Figure 1 and Appendix A).

To explore the phylogenetic relationships of the *BsSWEET* gene family, we constructed a phylogenetic tree with 71 genes from *B. striata*, *A. thaliana*, *Oryza sativa*, and *Vitis vinifera*. The results revealed a more intimate relationship between *B. striata* and *V. vinifera* (Figure 2). In accordance with previous studies, *SWEET* members in the abovementioned four species were divided into four subfamilies. The distribution of *BsSWEET* members in clades II and III was the largest, with nine (39.1%) and eight (34.8%), respectively. There were four and two *BsSWEET* genes in clades I and II, respectively. Based on the distribution and corresponding functions of the *SWEET* subfamily in *A. thaliana* and *O. sativa*, the potential functions of each *BsSWEET* gene member could be predicted. Plant phylogenetic analysis indicated that clade I primarily transported glucose, clade II transported hexose, clade III mainly transported sucrose, and clade IV transported fructose. It was speculated that *BsSWEET6*/*15*/*16* and other genes might be involved in the transport of sucrose in the plastid extracellular pathway [34].

### 2.3. Gene Structures and Conserved Motifs of BsSWEET Genes

The structural composition of genes can reflect their complexity and diversity of corresponding functions. To further understand the structural features of *BsSWEETs*, we analyzed the introns and exons of each member (Figure 3). The exon number of the entire *BsSWEET* gene family ranged from two to six, and within the same subfamily, the gene lengths and exon numbers were similar. For example, the *BsSWEET* genes were the shortest in clade II, with three to four exons, whereas the genes of clade IV were longer with five exons. A factor that may have been related to gene duplication and random insertion is the similar gene structures of the same subfamily, and the different gene structures of different subfamilies. Furthermore, we analyzed the motif distribution patterns of *BsSWEETs* and found that there were eight conserved motifs in BsSWEET proteins (Appendix A and Appendix A), among which motif1 and motif2 were the most widely distributed and belonged to the conserved MtN3/saliva domain. Moreover, members of the same subfamily shared similar motifs in number and composition, which also demonstrated the reliability of the phylogeny.

### 2.4. Cis-Element Analysis in Promoters of BsSWEET Genes

To investigate the potential functions of the 23 *BsSWEET* genes, we performed *cis*-acting element analysis in the promoter regions of *BsSWEETs*, focusing on the response elements related to sugar and stress responses (Figure 4 and Appendix A). There were 16 major *cis*-elements in *BsSWEETs* including: six sugar-response elements (e.g., G-box, I-box, and W box) [35,36,37]; five phytohormone-response elements (e.g., ABRE (Abscisic acid responsiveness), P-box (Gibberellin responsiveness), and TCA-Element (Salicylic acid responsiveness)); two light-response elements (e.g., ACE and TCT-Motif); two stress-response elements (e.g., LTR (low temperature responsiveness) and TC-rich repeats (defense and stress responsiveness)); and one development-related *cis*-acting element, the CAT-box (related to Meristem expression). *BsSWEET11* and *BsSWEET22* contained the most sugar-response elements, which also reflected that these two members might play essential roles in sugar regulation. The above results also indicated that the *SWEET* genes were functionally diverse and involved in different biological processes, including sugar transport, plant hormone responses, multiple abiotic stress responses, and growth and development.

### 2.5. Synteny and ω Analysis of BsSWEET Genes

A total of 21 *BsSWEETs* were unevenly distributed on 11 chromosomes, and 2 genes were not anchored on chromosomes (Figure 5A). Among them, Chr4 contained the most *BsSWEET* genes (six genes, accounting for 26.1%), Chr1, Chr5, Chr7, Chr8, and Chr11 contained two *BsSWEET* genes, while Chr2, Chr3, Chr6, Chr9, and Chr10 contained only one *BsSWEET* gene.

A total of 23 *BsSWEETs* genes were involved in six fragment replication events (*BsSWEET2*/*17*, *BsSWEET3*/*18*, *BsSWEET4*/*9*, *BsSWEET6*/*16*, *BsSWEET13*/*18*, and *BsSWEET15*/*16*) and two tandem replication events (*BsSWEET7*/*8* and *BsSWEET12*/*13*) (Figure 5B). These results revealed that gene duplication was the main driver of the *SWEET* gene family expansion in *B. striata*.

To explore the evolutionary relationships of *BsSWEETs* between species and the retention and loss of homologous genes, we developed a collinearity map between *B. striata*, *A. thaliana*, *V. vinifera*, and *Vanilla planifolia* (Figure 5C). There were 6, 10, and 20 pairs of homologous genes between *B. striata* and *A. thaliana*, *V. vinifera*, and *V. planifolia*, respectively. The results indicated that *B. striata* and *V. planifolia* had higher homology and were more closely related to the phylogeny and evolution of monocotyledonous species.

Further, to understand the evolutionary constraints acting on *SWEET* gene members, we calculated the nonsynonymous nucleotide substitutions ω ratio in *B. striata*, and estimated the variable ratio between codons in *BsSWEET* genes (Appendix A). The *SWEET* genes were evolved in a neutral (ω = 1) manner or under purifying selection (ω < 1), which suggested that they may have undergone strong purifying selection during evolution and were not affected by the external environment; thus, they were more functionally conserved.

### 2.6. Tissue-Specific Expression Patterns of BsSWEETs

The expression results indicated that *BsSWEETs* were differentially expressed in various tissues (Figure 6). Among them, nine *BsSWEET* genes were highly expressed in roots and 14 *BsSWEET* genes were highly expressed in flowers, which suggested that these genes may play a critical role in reproductive development. Further, two *BsSWEET* genes were highly expressed in stems. The *BsSWEET* genes were all negligibly expressed in leaves, which indicated that they may be involved in the growth and development of source organs in *B. striata* and mediate the unloading of sucrose in the extracellular pathway.

### 2.7. Expression Patterns of BsSWEETs in Responses to Low Temperature and Oxidative Stress

To assess the expression profile in responses to low temperature and oxidative stress, six *BsSWEET* genes from each subfamily were randomly selected based on available phylogenetic distance. The results revealed that the response patterns of the various genes to different treatments were inconsistent (Figure 7 and Figure 8). For low temperature treatment (4 °C), the expression levels of *BsSWEET1* and *BsSWEET10* were largely unchanged at 6 h post-treatment, but significantly up-regulated at 12 h post-treatment; *BsSWEET8* was significantly down-regulated; *BsSWEET14* expression was initially down-regulated and then up-regulated; *BsSWEET18* expression was most pronounced at 12 h, with a 60-fold up-regulation. For oxidative treatment (Cu_2_SO_4_), *BsSWEET1*/*10*/*17*/*18* were all up-regulated, with *BsSWEET18* up-regulated nearly 30-fold. In conclusion, both *BsSWEET10* and *BsSWEET18* can be considered as candidate genes to explore the response of *BsSWEETs* to low temperature and oxidative stress responses.

### 2.8. The Subcellular Localization of BsSWEET15 and BsSWEET16

The subcellular localization of the BsSWEET protein was investigated by transiently expressing BsSWEET15-GFP and BsSWEET16-GFP fusion proteins in *A. thaliana* protoplasts and observing the localization of GFP (green fluorescent protein) by laser confocal microscopy. The results revealed that the fluorescence of 35S: GFP control was distributed throughout the whole cell, whereas those of BsSWEET15 and BsSWEET16 were exclusively localized on the cell membrane (Figure 9). They were both membrane-localized proteins that may serve as carriers to transport sugars into or out of cells.

### 2.9. Protein Interaction Analysis of BsSWEETs

Orthoven2 was initially used to explore the protein interaction relationships of SWEETs, and conduct a homology comparison, where *AtSWEET12* was a homologous gene of *BsSWEET15*. Subsequently, the protein interaction network of AtSWEET12 was constructed on the string website, which had a strong interaction with a sucrose transporter, *AtSUC2*. Homologous to *AtSUC2, CTG2800.4* was found to be one of the members of the *SUT* gene family in *B. striata* [38]. Thus, we speculated that *BsSWEET15* and *CTG2800.4* might also have protein interactions (Figure 10). Based on the above interacting proteins, we preliminarily concluded that *BsSWEET15* might be involved in the regulation of sugar transport, plant growth, and development.

## 3. Discussion

*B. striata* is one of the most significant medicinal and ornamental Orchidaceae worldwide. SWEETs and STPs are important regulators in sugar metabolism pathways. The identification of the *SWEET* gene family in *B. striata* can lay a foundation for exploring its molecular mechanisms in plant growth, development, and defense processes.

There are 23 *SWEET* genes in *B. striata*. The *SWEET* gene family has been found in the following numbers: 17 in *A. thaliana*, 21 in *O. sativa*, 15 in *V. vinifera*, 29 in *Solanum lycopersicum*, and 24 in *Zea mays* [39,40]. The genome sizes of *A. thaliana*, *O. sativa*, *V. vinifera*, *S. lycopersicum*, and *Z. mays* were 125 Mb, 466 Mb, 487 Mb, 900 Mb, and 2.3 Gb, respectively [41,42,43,44,45]. Therefore, the number of *SWEET* genes in plants is independent of genome size, but correlated with gene replication events. A total of 23 *BsSWEET* genes were involved in six fragment replication events and two tandem replication events, which were the main drivers of *SWEET* gene family expansion in *B. striata*. Those evolved in a neutral (ω = 1) manner or under purifying selection (ω < 1) suggested that they may have undergone strong purifying selection during evolution and were not influenced by the external environment. Thus, *BsSWEET* genes were more functionally conserved. Expansions of *SWEET* genes have also occurred in other species, such as *JrSWEET* genes [46].

According to phylogenetic analysis, the SWEETs were classified into four subfamilies. SWEET proteins were predicted to harbor two MtN3/saliva domains in monocots and dicots. Most *BsSWEET* genes contain two MtN3/saliva domains; the others contain one MtN3/saliva domain. This is consistent with the comparative sequence analysis of the *ClaSWEET* genes [47]. It was found that the gene structures and protein structural domains of each subfamily were arranged similarly, indicating structural and functional conservation among the same subfamilies. However, there were differences among subfamilies, which may be related to functional diversity. *AtSWEET1* (clade I) serves as a unidirectional low-affinity glucose transporter that regulates glucose transport [48]. *OsSWEET5* (clade II) has galactose transport activity [49]. *AtSWEET11* and *AtSWEET12* (clade III) transport sucrose from leaves into vascular bundles [20]. *AtSWEET16* and *AtSWEET17* (clade IV) are responsible for fructose transport [50]. Our study provides a reference for further investigations into the functions of *BsSWEET*. It is worth mentioning that the *SWEET* gene family has now been identified in *Dendrobium officinale*, which suggests that *DoSWEET* genes may be associated with fleshy stems and medicinal compounds such as polysaccharides [51]. However, the *SWEET* gene family has not been systematically and completely analyzed in Orchidaceae. Our initial comprehensive analysis of the *SWEET* gene family in *B. striata* will provide new insights into *SWEET* genes related to sugar transport, growth, and development in medicinal Orchidaceae.

To gain underlying functional and regulatory mechanisms, we investigated its *cis*-acting elements. Five types of *cis*-acting elements were identified, including sugar-responsive, light-responsive, phytohormone-responsive, stress-responsive, and development-related, indicating that *BsSWEETs* genes were functionally diverse and might respond to different biological processes. Twelve *cis*-elements related to sugar repression and seven *cis*-elements related to sugar induction were identified in the *ZjSWEET2.2* promoter; *ZjSWEET*2.2 was confirmed as an export sugar from photosynthetic leaf cells with their gene expressions being regulated by sugar signals [52]. In this study, *BsSWEET11*/*22* contained the most sugar-responsive elements (six), implying that those likely played vital roles in sugar regulation.

The results of subcellular localization revealed that *BsSWEET15*/*16* were localized in the plasma membrane, and the others were predicted to be localized in the plasma membrane. It was shown that SWEET proteins could homo- and heterooligomerize to generate functional pores and provide channels. *SlSWEET7a*/*14* localized in the plasma membrane in *S. lycopersicum*, formed homodimers and heterodimers, and established pores to transport sugars, particularly for larger substrates such as sucrose and fructose [53]; thus, it was hypothesized that they served as carriers to transit sugars. The formation of the potato tuber is related to photoperiods and sugar transport, and involves *StSWEET11*, *SUTs*, and *StSP6A*, which alter sucrose accumulation and transport in plants. Proteins were linked by STRING, in which both SUC2 and SUT4 proteins were sucrose transporters. The homologue of *AtSWEET12* is *BsSWEET15*, whereas the homologue of *AtSUC2* is *CTG2800.4*; thus, it was speculated that *BsSWEET15* and *CTG2800.4* may interact to participate in sucrose transport in *B. striata*. In addition, *AtSWEET13* and *AtSWEET14* signaling occurred during anther dehiscence and germination-mediated GA, which indicated that these transporters may be required for different stamen development processes [54]. q-PCR (Quantitative real time polymerase chain reaction) was employed to examine the expression patterns of *BsSWEETs* in different tissues, and the number of *BsSWEETs* expressed in flowers was higher than other tissues, which suggested that those might play a unique role in reproductive development.

*SWEET* genes are involved in abiotic stress responses and exhibit significant functional diversification [20]. *AtSWEET16* confers an increased tolerance to low temperatures [50]. *GhSWEET* genes might also enhance the adaptability of plants to diverse abiotic stresses [55]. To test the regulation of miR398 by heavy metal-induced oxidative stress, miR398 expression was down-regulated in *A. thaliana* seedlings under high Cu^2+^ and Fe^3+^ exposure. In this study, most *SWEETs* expression patterns were altered under stress conditions. *BsSWEET8* was down-regulated under low temperature and *BsSWEET14* was down-regulated under oxidative treatments. However, *BsSWEET10* and *BsSWEET18* were up-regulated under low temperature and oxidative treatments conditions, indicating that *BsSWEETs* are involved in stress resistance. The functional diversity of the *SWEET* gene family provides potential anti-stress genes for further germplasm screening and quality improvement in *B. striata*.

The identification and characterization of the *SWEET* gene family in *B. striata* can assist with further analyzing its regulatory mechanisms, while elucidating their critical biological functions. However, the biokinetics of *BsSWEET10*/*18*/*15*/*16* for abiotic stresses, plant growth, development, and sugar transport in *B. striata* warrant further detailed investigations.

## 4. Materials and Methods

### 4.1. Genome-Wide Identification, Protein Features, and Chromosomal Locations of BsSWEET Genes

The genomic and annotative data for *B. striata* was obtained from our laboratory. AtSWEET protein sequences were retrieved from the TAIR database (https://www.arabidopsis.org/, accessed on 12 March 2022); the *BsSWEET* gene was searched with BLAST and candidate family members were obtained. Further, HMM (Hidden Markov Model) mapping of the MtN3/saliva conserved domain (PF03083) was downloaded from the Pfam database (http://pfam.xfam.org/, accessed on 13 March 2022), which confirmed its presence in the candidate sequences, and was further confirmed using SMART (http://smart.embl.de/, accessed on 15 March 2022), InterPro (http://www.ebi.ac.uk/interpro/, accessed on 15 March 2022), and NCBI-CDD (https://www.ncbi.nlm.nih.gov/cdd/, accessed on 15 March 2022), respectively [56,57,58]. Lastly, the nucleotide and amino acid sequences of the 23 *BsSWEET* genes were verified. According to their positions on respective chromosomes, all *SWEET* genes were sequentially numbered, the chromosomal location of the *BsSWEET* genes was displayed by TBtools v1.089 (Chen, C., et al, China), and the chromosomal location of the above *BsSWEET* genes was derived from the genome annotation files. In addition, Expasy ProtParam (http://www.expasy.org/tools/protparam.html, accessed on 20 March 2022) and WoLF PSORT II (https://wolfpsort.hgc.jp/, accessed on 21 March 2022) were employed to predict the physicochemical characteristics and subcellular localization of the BsSWEET proteins, respectively [59,60].

SOPMA (http://npsa-pbil.ibcp.fr/cgi-bin/npsa_automat.pl?page=npsa_sopma.html, accessed on 24 March 2022) and Swiss-Model (https://swissmodel.expasy.org/, accessed on 24 March 2022) were used to predict the secondary and tertiary structures of the BsSWEET proteins, respectively [61,62]. Moreover, SignalP (http://www.cbs.dtu.dk/services/SignalP/, accessed on 26 March 2022), TMHMM v2.0 (http://www.cbs.dtu.dk/services/TMHMM/, accessed on 26 March 2022), and NetPhos v2.0 (http://www.labtools.us/netphos-2-0/, accessed on 27 March 2022) were utilized to predict and analyze the signal peptides, transmembrane domains, and phosphorylation sites of the BsSWEET proteins, respectively [63,64,65].

### 4.2. Phylogenetic Analysis of SWEET Genes

The multiple sequence alignment of BsSWEET core protein sequences in *B. striata* was performed using Clustal v1.83 (Higgins D.G., et al., Ireland) and Geneious v11.0 (Biomatters, Auckland, New Zealand) with default settings [66]. To investigate the phylogenetic relationships among different *SWEETs*, MEGA v7.0 (Mega Limited, Auckland, New Zealand) was used to construct a phylogenetic tree of SWEET proteins from *B. striata*, *A. thaliana, O. sativa*, and *V. vinifera* by the Neighbor-joining (NJ) method, and the bootstrap was repeated 1000 times [67]. Among them, the data for *O. sativa* and *V. vinifera* were obtained from NCBI. In addition, it was annotated and modified by EvolVIEW v2.0 (https://www.evolgenius.info/evolview-v2/, accessed on 30 March 2022) [68].

### 4.3. Analysis of Conserved Motifs, Gene Structures, and Cis-Acting Elements of the BsSWEET Genes

The conserved motifs of *BsSWEET* genes were predicted by MEME (https://meme-suite.org/meme/tools/meme, accessed on 5 April 2022), with eight searched conserved motifs and the others being default values [69]. The *BsSWEET* genes structures were visualized using TBtools v1.089 to identify exon–intron boundaries. Further, the Plant CARE database (http://bioinformatics.psb.ugent.be/webtools/plantcare/html/, accessed on 6 April 2022) was used to predict and analyze the *cis*-acting elements of the 2500 bp upstream sequence of each *BsSWEET* gene and visualized using TBtools v1.089 [70,71]. Finally, the exon–intron structures and 2500 bp upstream sequences of the *BsSWEET* genes were obtained from the genome annotation files.

### 4.4. Analysis of Gene Duplication and Synteny in B. striata

The *SWEET* genes homology analysis of *B. striata* was performed using MCScanX (Wang, Y., et al, China) and visualized by Circos (Krzywinski, M., et al, Canada) [72,73]. The homology of *B. striata* and three species of *A. thaliana*, *V. vinifera*, and *V. planifolia* were analyzed using MCScanX and Tbtools v1.089. The *V. planifolia* data were obtained from NCBI.

The ratios of non-synonymous (Ka) and synonymous (Ks) divergence levels were calculated by pairing the BsSWEET protein sequences in Clustal v1.83. The ω ratio of all *BsSWEET* sequences was estimated by the codeml program embedded in the PAML package and their molecular selection effects were assessed [74,75].

### 4.5. Plant Materials and Treatments

*B. striata* was planted in a greenhouse at the National Engineering Laboratory for Resource Development of Endangered Crude Drugs in Northwest China. The *B. striata* seedlings under similar growth conditions were transplanted into square plastic flowerpots (7 cm long × 7 cm wide × 8 cm high, four plants per pot) containing humus and sand (1:2 ratio), and placed in a light incubator at 30 °C, under 55% relative humidity and a 16 h light/8 h dark cycle. The roots, stems, leaves, and flowers of well-grown *B. striata* were gathered for tissue-specific expression profiling.

Two-month-old seedlings were selected for the stress treatment experiments. To determine the low temperature responses of *B. striata* plants, they were placed in a 4 °C incubator and whole plants were gathered at 0, 1, 3, 6, and 12 h after treatment, respectively. For oxidative stress, a 100 μM Cu_2_SO_4_ solution was sprayed on whole *B. striata* plants, which were gathered at 0, 1, 3, 6, and 12 h following treatment, respectively. All samples were immediately frozen and stored in a −80 °C refrigerator and three biological replicates were used.

*A. thaliana* seedlings (Col-0 ecotype) under similar growth conditions were transplanted into square plastic pots (7 cm long × 7 cm wide × 8 cm high, two plants per pot) containing substrate soil, perlite, and vermiculite (3:1:1), and cultured at 22 °C, 55% relative humidity, and 8 h light/16 h dark. Subcellular localization experiments were conducted using healthy *A. thaliana* with no shoot materials.

### 4.6. RNA Extraction and q-PCR Analysis

The total RNA was extracted from each sample using the Polysaccharide and Polyphenol Plant Rapid RNA Isolation Kit (TaKaRa, Dalian, China), after which the concentration and quality of RNA were measured with a NanoDrop 2000c Spectrophotometer (Thermo Scientific, USA). The cDNA was synthesized from 1 μg of total RNA using the HiScript II Q Select Reverse Transcriptase Kit (Vazyme Biotech, Nanjing, China).

The q-PCR primers were designed with genscript (https://www.genscript.com/tools/real-time-pcr-taqman-primer-design-tool, accessed on 8 April 2022), and q-PCR was performed on a LightCycler 96 system (Roche Diagnostics GmbH, Mannheim, Germany) with SYBR Green q-PCR Master MIX (Vazyme, Nanjing, China) (20 μL reaction solution, including 10 μL MasterMix, 0.4 μL forward primer, 0.4 μL reverse primer, 0.5 μL template cDNA, and 8.5 μL sterile deionized water). Three technological replications were used for each reaction, and the *BsGAPDH* gene was used as the internal reference gene. The reaction procedure was as follows: 95 °C for 1 min; 95 °C for 5 s; 58 °C for 30 s; 72 °C for 30 s, for a total of 45 cycles. The relative expressions of *BsSWEETs* were calculated using the 2^−ΔΔCT^ method [76]. Finally, the data were analyzed via DPS v9.01 (China) and Microsoft office Excel v2019 (Microsoft, Redmond, MA, USA), and curves were drawn by TBtools v1.089 and GraphPad v8.0 (San Diego, CA, USA), with all q-PCR primers listed in Appendix A. Duncan’s test was employed to compare significant differences between treatments, and all data were the mean ± standard error of three biological and three technical replicates.

### 4.7. Expression Patterns of BsSWEET Genes

Using the q-PCR data for roots, stems, leaves, and flowers, the expression patterns of *BsSWEET* in different tissues were analyzed to explore the functions of *BsSWEETs* in tissue development, and heatmap was created with TBtools v1.089.

### 4.8. The Subcellular Localization of BsSWEET15 and BsSWEET16

We randomly selected *BsSWEET15* and *BsSWEET16* to perform subcellular localization studies, and in plant cells, the ORF of B*sSWEET15* and *BsSWEET16* with no terminator codon was reamplified with ORF-F and ORF-R primers harboring Kpn I and BamH I sites. The full-length sequences of B*sSWEET15* and *BsSWEET16* were cloned into the pHBT-GFP-NOS vector. Among them, the primers used for ORF amplification were designed by Primer Premier 5, as listed in Appendix A. Further, healthy *A. thaliana* leaves were selected to prepare protoplasts, which transformed the confirmed recombinant vector [77]. Transformed protoplasts were typically incubated at 22 °C for 16 h in the dark. The empty vector expressing untargeted GFP was used as a control. GFP fluorescence images were captured using a high-resolution confocal laser microscope (Leica TCS SP5, LEICA, Wetzlar, Germany) with a dual-channel red-green light scanning mode, (488 nm to excite GFP green fluorescence, and 543 nm to excite chloroplast autofluorescence).

### 4.9. Interactive Network Analysis of BsSWEET Genes

Ortho venn (http://www.bioinfogenome.net/OrthoVenn/, accessed on 15 April 2022) was used to identify orthologous pairs between *BsSWEETs* and *AtSWEETs* [78]. Next, based on an *A. thaliana* association model [79], the STRING database (http://string-db.org/cgi, accessed on 15 April 2022) was used to construct the predicted interaction network of *SWEET* genes in *B. striata*.

## Figures and Tables

**Figure 1 ijms-23-10057-f001:**
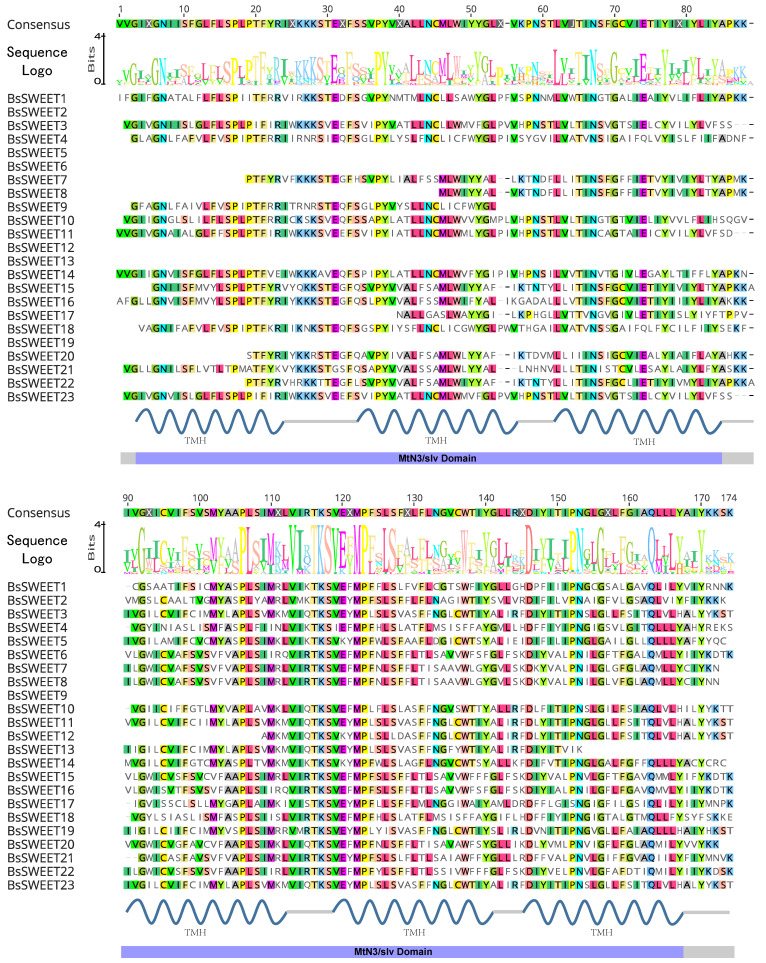
Multiple sequence alignments of BsSWEET proteins.

**Figure 2 ijms-23-10057-f002:**
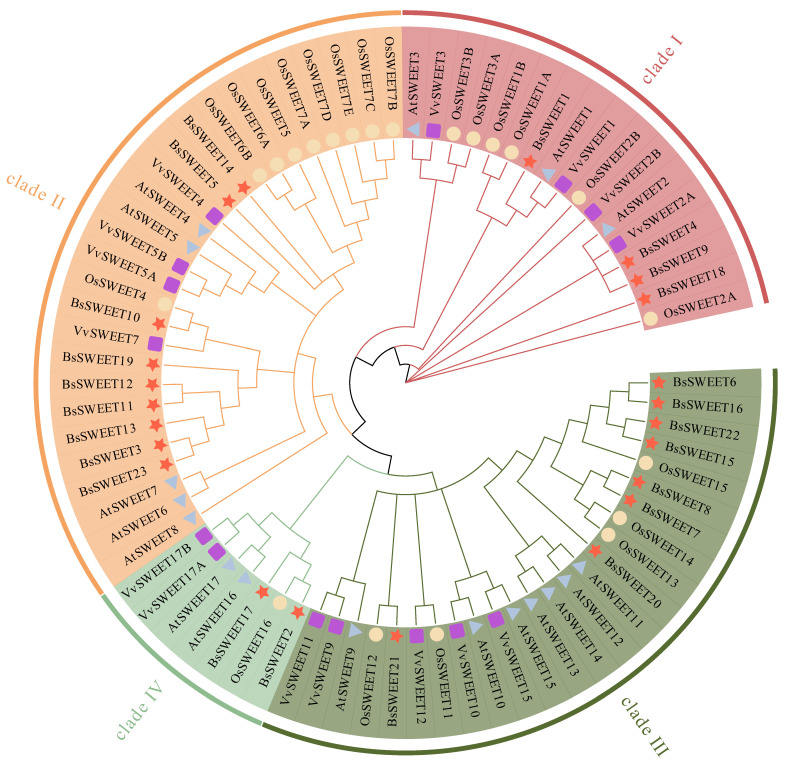
Phylogenetic relationships of SWEET proteins. Red, orange, dark green, and green indicate different clades (clade I, clade II, clade III, and clade IV) of SWEET genes. Red stars represent genes from *B*. *striata*. Blue triangles, yellow circles, and purple rectangles represent genes from *A. thaliana*, *O. sativa*, and *V. vinifera*, respectively. At, Os, Vv, and Bs represent *A. thaliana*, *O. sativa*, *V. vinifera*, and *B*. *striata*, respectively.

**Figure 3 ijms-23-10057-f003:**
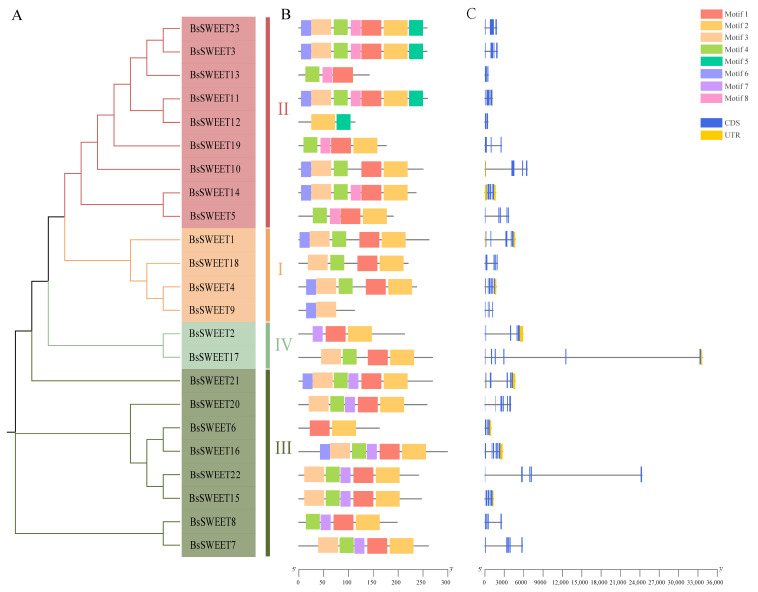
Phylogenetic relationships, gene structures, and architectures of conserved protein motifs in *BsSWEET* genes. (**A**) Phylogenetic relationship analysis of *BsSWEET* genes. Red, orange, dark green, and green indicate clade I, clade II, clade III, and clade IV, respectively. (**B**) Motif analysis of BsSWEET proteins. Colored boxes numbered 1–8 indicate different motifs. Detailed information for each motif is provided in Appendix A and Appendix A. The protein lengths can be estimated by the scale at the bottom. (**C**) Exon–intron structures of *BsSWEET* genes. CDS, UTR, and introns are indicated by blue, gold, and black lines, respectively. The lengths of exons and introns from each *BsSWEET* gene are presented proportionally.

**Figure 4 ijms-23-10057-f004:**
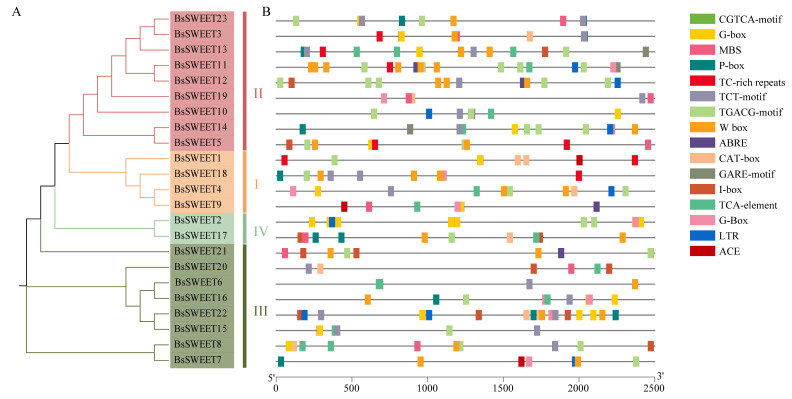
Phylogenetic relationships and *cis*-acting elements in *BsSWEET* genes. (**A**) Phylogenetic relationship analysis of BsSWEET proteins. Red, orange, dark green, and green indicate clade I, clade II, clade III, and clade IV, respectively. (**B**) Functions of *cis*-acting elements found in the promoter regions of the *BsSWEET* genes. The promoter sequences of 23 *SWEET* genes (−2500 bp) were analyzed by PlantCARE. The upstream lengths of the translation start sites can be inferred from the scale at the bottom.

**Figure 5 ijms-23-10057-f005:**
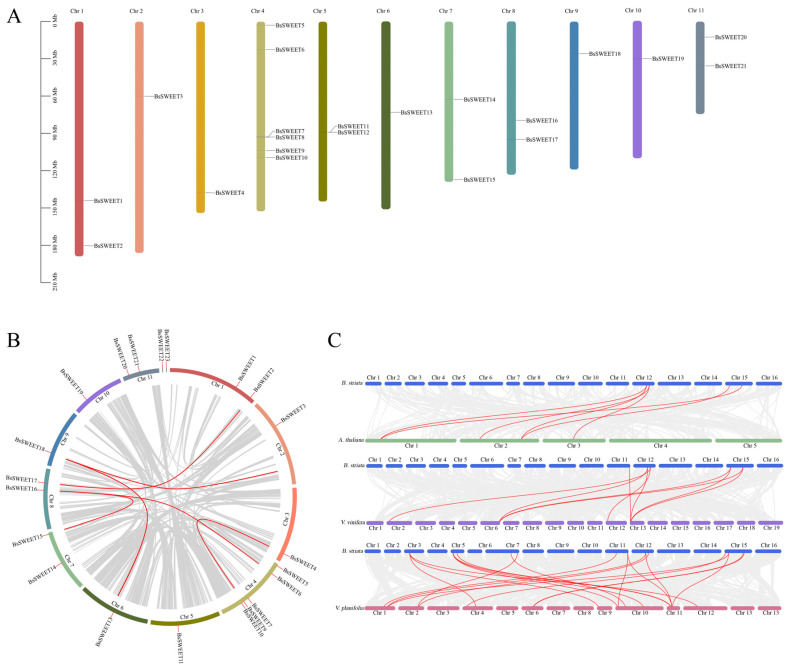
Repeat and synteny analysis. (**A**) Chromosomal distribution of *BsSWEET* genes. The chromosome numbers and sizes are indicated at the top and left of each bar. (**B**) Duplication analysis of *BsSWEET* genes. (**C**) Covariance analysis of *SWEET* genes between *B. striata* and other plants. Red lines represent homologous gene pairs.

**Figure 6 ijms-23-10057-f006:**
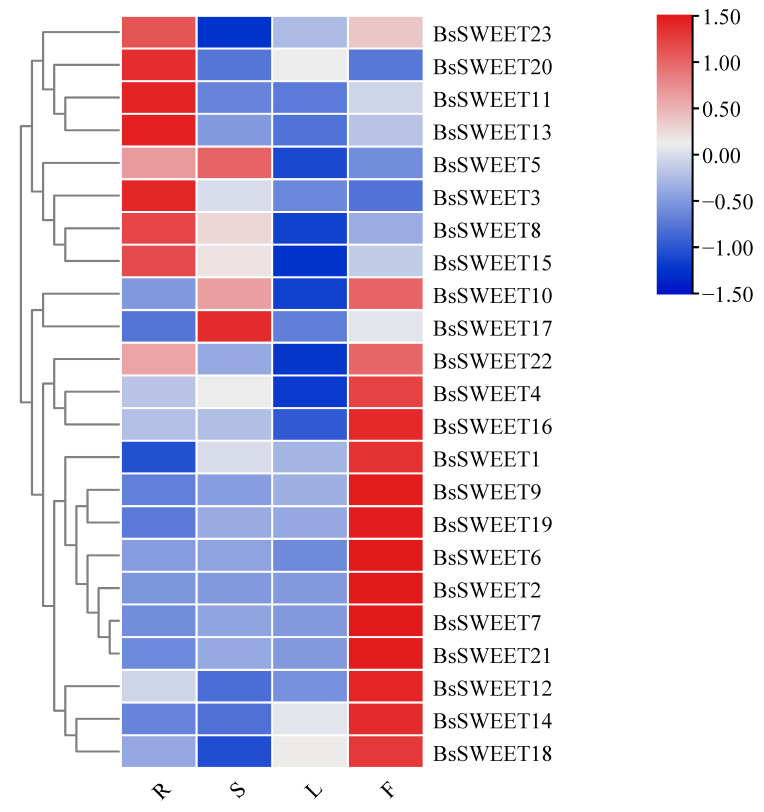
Expression profiles of *BsSWEET* genes in different tissues. Expression levels of 23 *BsSWEETs* were analyzed by q-PCR in the roots, stems, leaves, and flowers. A heatmap was constructed by log_2_-transformed expression levels. R, S, L, and F represent root, stem, leaf, and flower, respectively.

**Figure 7 ijms-23-10057-f007:**
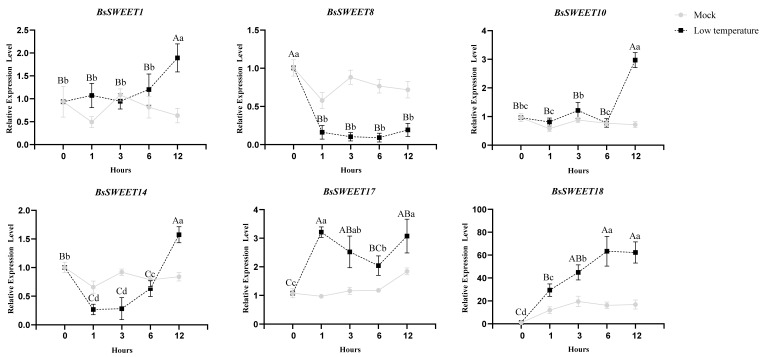
Relative expression levels of *BsSWEETs* under simulated mock control, low temperature treatments. All data represent averages of three biological replicates, error bars indicate SD. *p* < 0.05 (lowercase letters) and *p* < 0.01 (capital letters) were considered statistically significant by the Duncan’s test.

**Figure 8 ijms-23-10057-f008:**
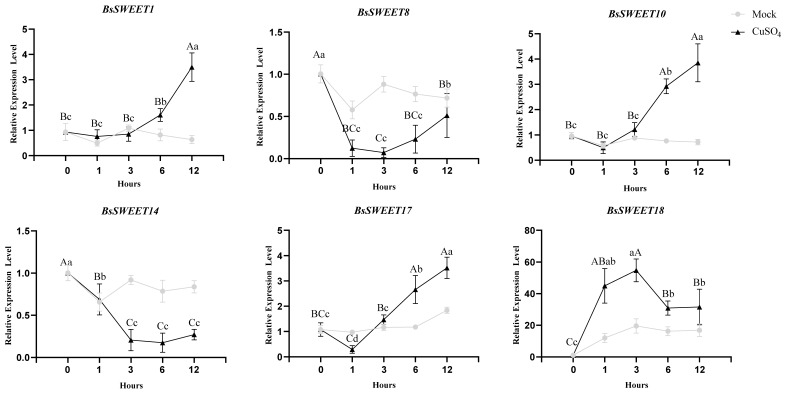
Relative expression levels of *BsSWEETs* under simulated mock control and Cu_2_SO_4_ solution treatments. All data represent averages of three biological replicates, error bars indicate SD. *p* < 0.05 (lowercase letters) and *p* < 0.01 (capital letters) were considered statistically significant by the Duncan’s test.

**Figure 9 ijms-23-10057-f009:**
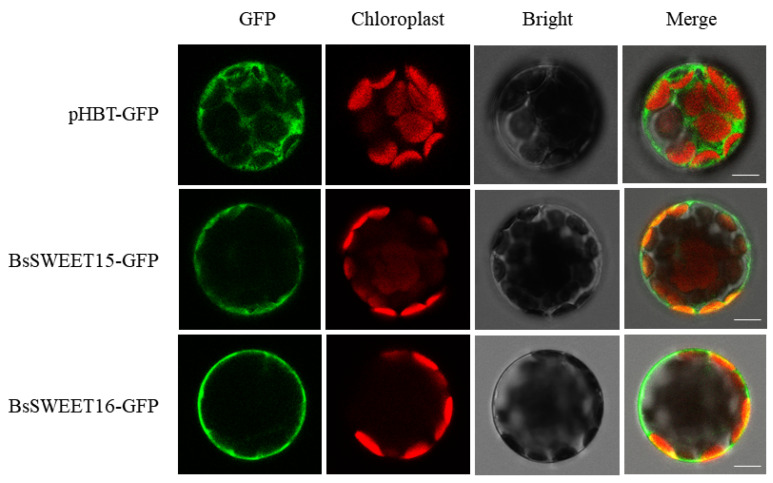
The subcellular localization of *BsSWEET15* and *BsSWEET16* in *A. thaliana* protoplasts. (Scale bar: 5 μm).

**Figure 10 ijms-23-10057-f010:**
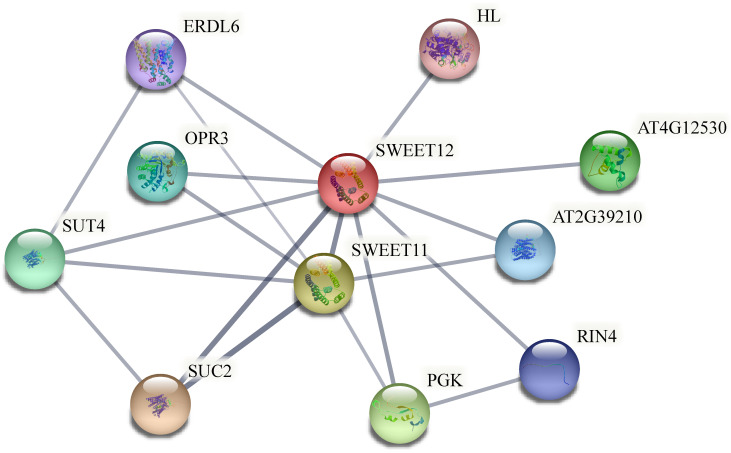
Functional interactive network of BsSWEET15 and its interacting proteins.

## Data Availability

MDPI Research Data Policies.

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
