# Peer review of "Genome-Wide Identification and Expression Patterns of the SWEET Gene Family in Bletilla striata and its Responses to Low Temperature and Oxidative Stress"

_ijms, 2022, doi:10.3390/ijms231710057_

Round 1

Reviewer 1 Report

" SWEET (Sugars will eventually be exported transporter) is a newly identified class of sugar transporters" -  SWEET is a very well known class of transporter proteins, this is not something "newly identified". " Further, BsSWEET10 and BsSWEET18 were highly expressed, which were important candidate genes for  studying the responses of B. striata to environmental stress. In addition, subcellular localization results indicated that BsSWEET15 and BsSWEET16 were localized in the cell membrane." - This is very weird, authors considered SWEET10 and 18 as important candidates but studied SWEET15 and 16 for localization study. The basis to select a few SWEETs for qPCR analysis is not clear. Selection based on phylogenetic position does not make any sense.  Authors can look into SWEET, SEMI-SWEET, or EXTRA-SWEET.  Fig 6 has the information provided in Fig 1. 

Reviewer 2 Report

The authors provided a quality article. It is well written and framed. The only thing that is not entirely clear is why the authors used the Student’s t-test? Why not Duncan's test, or another ANOVA? I think it would give more information. In general, I think that the article of the authors deserves publication in the journal.

Reviewer 4 Report

Main comments:

The title of the manuscript appears to be very interesting. It would certainly arouse interest among breeders and scientists dealing with plant stresses and valuable plant that seems to be Bletilla striata. However, the manuscript needs to be thoroughly revised, as spraying plants with copper sulphate (Line 450 in Materials and Methods) cannot be considered as inducing salt stress. Application of Cu2SO4 to leaves can cause oxidative stress and leaf burn. Copper sulphate is used in organic plant cultivation as an agent against fungal plant diseases. In addition, the introduction lacks a description and review of the literature on the sensitivity of Bletilla striata to chill temperatures and possibly to the action of copper sulphate as an antifungal agent and not as a salt stress agent. For this reason, the manuscript cannot be published in its current version.

Detailed comments and suggestion:

The title

Line 2-4: The title does not fully correspond to the content of the manuscript. “Abiotic stresses” it is too broad a term to be used in the description of the research.

Abstract: Line 23: rather, it is about chill stress (4°C) and no salinity - spraying with copper sulfate.

Introduction: there is no description of Bletilla striata reaction to cold stress and copper sulphate applications.

Results: Line 251: chill stress and Cu2SO4 treatment (application)

The manuscript is well written, supported by well-prepared numerous figures and photos. And its publication, as amended, may be considered.

Round 2

Reviewer 1 Report

The authors have tried to address all of my concerns. The revised version of the MS looks appropriate.

Reviewer 4 Report

The manuscript has been completed and revised in line with the comments and suggestions provided in the review.